# The Impacts of Soil Moisture Initialization on the Forecasts of Weather Research and Forecasting Model: A Case Study in Xinjiang, China

**Hailiang Zhang [1][ID], Junjian Liu [1], Huoqing Li [1], Xianyong Meng [2,\*][ID] and Ablimitijan Ablikim [3]**

[1] Institute of Desert Meteorology, China Meteorological Administration, Urumqi 830002, China; zhanghl@idm.cn (H.Z.); liujj@idm.cn (J.L.); lihq@idm.cn (H.L.)
[2] College of Resources and Environmental Science, China Agricultural University, Beijing 100094, China
[3] Xinjiang Meteorological Observatory, Urumqi 830002, China; ablimit128@163.com
\* Correspondence: xymeng@cau.edu.cn

**Abstract:** Soil moisture is a critical parameter in numerical weather prediction (NWP) models because it plays a fundamental role in the exchange of water and energy cycles between the atmosphere and the land surface through evaporation. To improve the forecast skills of the Weather Research and Forecasting (WRF) model in Xinjiang, China, this study investigated the impacts of soil moisture initialization on the WRF forecasts by performing a series of simulations. A group of simulations was conducted using the single-column model (SCM) from 1200 UTC on 15 to 18 August 2019, at Urumchi, Xinjiang (43.78° N, 87.6° E); another was performed using the WRF model for a real weather case in Xinjiang from 0000 UTC 15 August to 1200 UTC 18 August 2019, which included an episode of heavy precipitation and gales. Our most notable findings are as follows. Specific humidity increases and potential temperature decreases persistently when soil moisture increases because of soil water evaporation. Soil moisture initialization could impact the energy budget and modulate the partition of the total available energy at the land surface significantly through evaporation and the greenhouse effect. Replacing the soil moisture with a proper multiple of the National Centers for Environmental Prediction (NCEP) Global Forecast System (GFS) soil moisture data could significantly improve the critical success index (CSI) and frequency bias (FBIAS) of precipitation and the root-mean-squared errors (RMSEs) of 2-m specific humidity and 2-m temperature. These findings indicate the prospect of a new way to improve the forecast skills of WRF in Xinjiang or other similar regions.

**Keywords:** soil moisture; WRF; evaporation; precipitation; land–atmosphere interaction; energy budget

## 1. Introduction

Soil moisture affects the planetary boundary layer's (PBL) evolution through evaporation and sensible heat fluxes. This results in differences in the mesoscale vertical circulation [1,2]. A higher soil moisture typically causes lower soil and air temperatures, more stable and shallower boundary layers (BL) and higher humidity and moist static energy in the BL. This leads to a lower cumulus cloud base and higher convective available potential energy [3]. Soil moisture can significantly influence the heat and water exchanges between the atmosphere and the land surface [4,5]. Therefore, it plays some important roles in numerical weather prediction (NWP) models.

Since soil moisture is critical for weather prediction, numerous sensitivity studies investigating the impacts of soil moisture initialization on the NWP have been conducted. These studies suggested that: (1) soil moisture is the most important controlling parameter of surface energy fluxes and

budget [6–9]; (2) soil moisture significantly affects the variations in temperature and precipitation when soil moisture–atmosphere interactions are strong [10]; (3) the variations in surface temperature and water vapor induced by soil moisture can alter the structure of the atmospheric boundary layer and cause the formation of shallow clouds [11–15]; (4) compared to the initial soil moisture, soil moisture evolution has little impact on the mean near-surface thermodynamic variables [16]. In a sense, soil moisture initialization plays a key role in NWP simulations and can influence the forecasts of the NWP through the variations in surface temperature and water vapor that are induced by the soil moisture. To improve the predictive skills of NWP, many studies have been conducted using different types of soil moisture data. Based on a Weather Research and Forecasting (WRF) model coupled with the Noah land surface model (LSM), Hong et al. [9] incorporated the High-Resolution Land Data Assimilation System (National Center for Atmospheric Research (NCAR), Boulder, CO, USA) (HRLDAS; developed to provide soil moisture data in high spatial resolution by NCAR) to improve soil moisture initialization. Zhong et al. [17] utilized HRLDAS to provide an alternative soil moisture initialization method and revealed that a drier soil moisture could lead to a noticeable change in energy partitioning at the land surface, which could improve the prediction of the diurnal 2-m temperature range. However, the improper initialization of soil moisture causes near-surface temperature and humidity forecast errors [18,19].

Previous studies have mostly focused on the land–atmosphere interaction driven by soil moisture or the impacts on the simulations of NWP caused by soil moisture initialization provided by the Global Land Data Assimilation System (GLDAS, (National Aeronautics and Space Administration (NASA), Washington, DC, USA)) or other reanalysis data [20–22]. The impacts of soil moisture initialization have rarely been investigated under relatively extreme conditions (e.g., the soil moisture content is of 3.0 multiple of the National Centers for Environmental Prediction (NCEP) Global Forecast System (GFS) data). Comprehensive studies on the impacts of soil moisture initialization on the forecasts of WRF were limited for the WRF model. To our knowledge, few studies have focused on (1) the impacts of soil moisture initialization on the forecasts of WRF in Xinjiang, which is the largest province in China, located in the arid region of Northwestern China; (2) how different (including relatively extreme) soil moisture initializations affect the forecasts of WRF simulations; and (3) the application of the single-column model (SCM) to isolate the simulations from the interference of large-scale advections in order to obtain reliable results of soil moisture's impacts on forecasts.

To apply the WRF model efficiently and improve the accuracy of forecasts of WRF in Xinjiang, it is necessary to rigorously examine the soil moisture's impacts on the forecasts of WRF. Firstly, we conduct SCM simulations to reveal the impacts of soil moisture initialization on the specific humidity and potential temperature in the boundary layer. Secondly, we evaluate how different soil moisture initializations can affect the forecasts of WRF simulations. Thirdly, we consider the upper air forecasts and the surface forecasts in order to comprehensively evaluate the impacts of soil moisture on the WRF forecasts. The study would reveal the impact of soil moisture initialization on the energy budget at the land surface through evaporation and greenhouse effects. It would also show how the predictive skills of WRF could be enhanced by replacing the soil moisture with a proper multiple of the NCEP GFS soil moisture data. These could provide prior knowledge and help to improve the forecasting performance of WRF in Xinjiang, China or other similar regions.

## 2. Data and Methods

### 2.1. Methods

As the SCM takes a single atmospheric column from a parent general circulation model (GCM), this column contains the physical parameterization schemes that are used in the GCM to represent unresolved subgrid-scale processes, while the model dynamics is replaced with boundary forcing [23]. Since the SCM eliminates three-dimensional dynamical core feedbacks, it is typically used under idealized forcing scenarios to develop a fundamental understanding about the atmosphere, such as

understanding the radiative–convective equilibrium [24]. SCMs have long been used for these purposes and remain popular among international communities today [25,26]. For example, the Global Energy and Water Cycle Experiment Cloud Systems study recommended SCMs as a key tool for modeling and understanding cloud systems [27,28]. Consistently, we employ the SCM model to study the impacts of soil moisture initialization on the atmosphere under the idealized forcing of soil moisture values ranging from 0.1 to 0.6 (i.e., 10–60%).

However, the SCM balance can easily drift away from a realistic atmosphere state, leading to results that might no longer be representative of the case under scrutiny. Since the interaction with the large-scale flow is impossible, the SCM might not always lead to identical results once implemented in the fully interactive global model [26]. To comprehensively investigate the impacts of soil moisture initialization on WRF forecasts, we evaluate the WRF's sensitivities with respect to the default and modified soil moisture levels in order to understand the role of soil moisture initialization on the land–air interactions in Xinjiang, China. The simulation period was from 0000 UTC 15 August to 1200 UTC 18 September 2019. During the study period, heavy precipitation and gale weather processes occurred in most regions of Northern Xinjiang, caused by the Middle Asia vortex (MAV). The detailed configurations of the SCM and WRF simulations are described in the next subsection.

## 2.2. Detailed Configuration and Data

To understand the impacts of soil moisture on the air specific humidity and potential temperature, four SCM simulations were conducted based on the simulated atmospheric profile and surface conditions (Table 1) on UTC1200 15 August for Urumchi, Xinjiang (43.78° N, 87.65° E), by replacing the initial soil moisture content with 0.1, 0.2, 0.4 and 0.6, respectively. Table 1 summarizes the initial conditions of the four SCM sensitivity tests. Parametrizations of the physical processes are applied in the SCM simulations by following the Yonsei University (YSU) scheme [29], the Noah land surface model (LSM) [6], the GCM version of the Rapid Radiative Transfer Model (RRTMG) [30] and the Revised MM5 Monin–Obukhov scheme. The simulation period started at 0000 UTC 15 August 2019 and ended at 0000 UTC 18 August 2019 (i.e., 72 h). To isolate the simulations from the interference of large-scale advections, this study excludes large-scale advective forcing in order to obtain reliable and robust results.

**Table 1.** Initial conditions of the single-column model (SCM) ideal experiment.

| Vertical Level | Model Top (km) | Soil Moisture (%) | U (m/s) | V (m/s) | Potential Temperature (K) | Specific Humidity (k/kg) | Coordinates (°) | Start Time (UTC) |
|---|---|---|---|---|---|---|---|---|
| 60 | 6 | 0.1 0.2 0.4 0.6 | 10 | −7 | 301(surface)–328(top) | 0 | 43.78° N 87.65° E | 2019081500 |

For the real weather case simulations (see Figure 1), the WRF configuration included two nested domains, with horizontal grid spacings of 9 km and 3 km, covering Xinjiang completely. Vertically, there were 50 hybrid sigma levels from the surface to the mode top set at 10 hPa. All simulations started at 0000 UTC 15 August 2019 and ended at 1200 UTC 18 August 2019 (i.e., 84 h). The initial and boundary conditions for pressure, water vapor, temperature and wind were obtained from the NCEP GFS data at 0.5° grid spacing and with 3-h time intervals.

To reveal the impacts of soil moisture on the WRF forecasts, six simulations were conducted. As limited-area mesoscale models, like the WRF, their soil moisture initial conditions are typically obtained from operational modeling systems such as the NCEP's GFS. The first WRF simulation was conducted without any update (i.e., control experiment), for which the initial soil moisture was provided by NCEP GFS data. In the remaining five simulations, the soil moisture was set to 1.25, 1.5, 2.0, 2.5 and 3.0 multiples of NCEP GFS soil moisture content, respectively, at each layer of the Noah

LSM. Figure 2 illustrates the initial conditions of soil moisture content of the 10–cm topsoil layer of the Noah LSM for six WRF simulations. The "1.0 soil moisture" represents the 1.0 multiple of NCEP GFS soil moisture, the "1.25 soil moisture" represents the 1.25 multiple of NCEP GFS soil moisture, and so on.

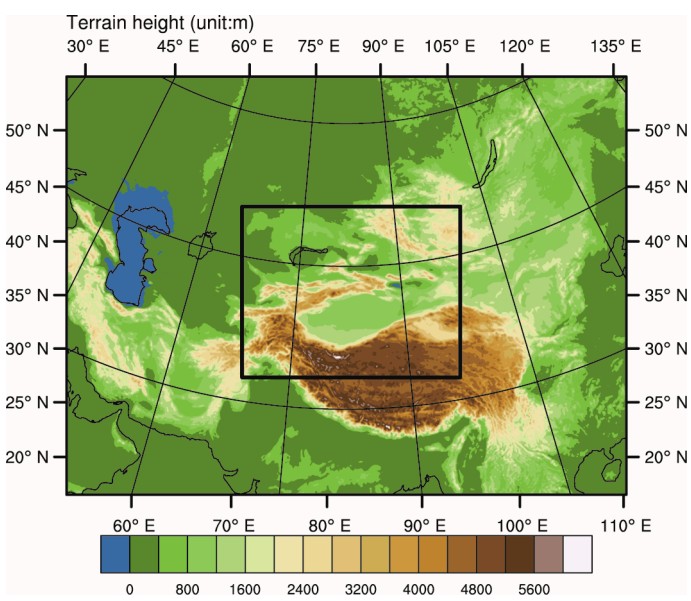

**Figure 1.** The two nested domains with horizontal grid spacings of 9 and 3 km for Weather Research and Forecasting (WRF) simulation. The inner simulation domain covers Xinjiang completely.

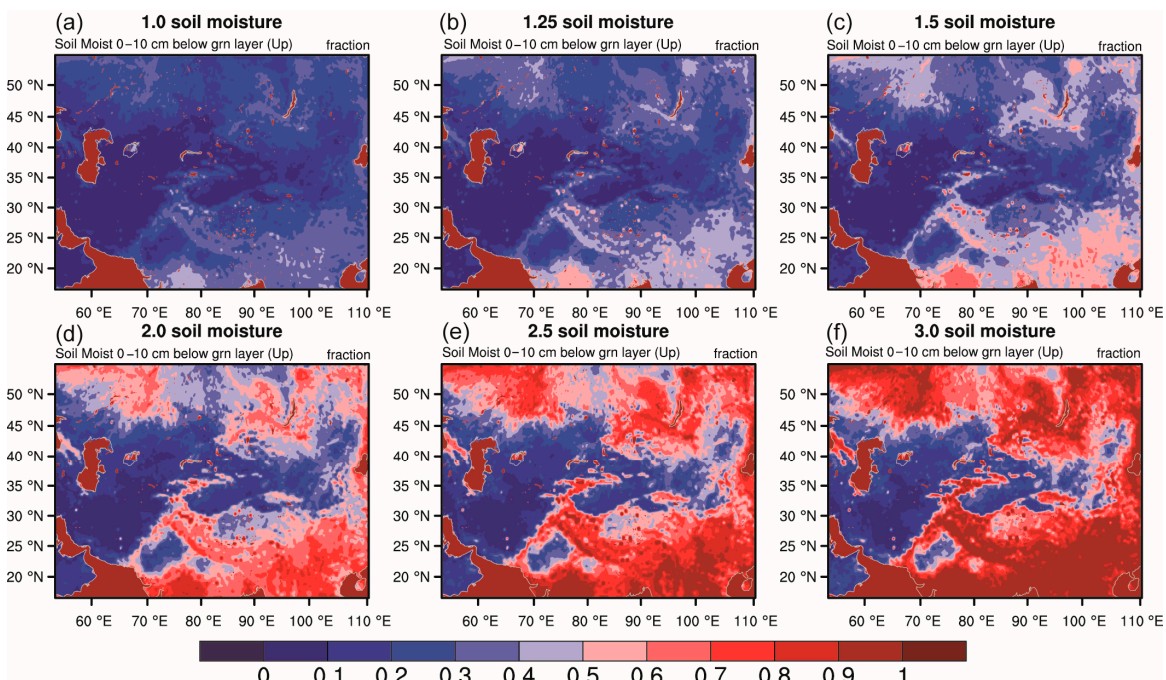

**Figure 2.** The soil moisture contents of 10–cm topsoil layer of the Noah land surface model (LSM) of six WRF simulations: (**a**) 1.0 multiple of National Centers for Environmental Prediction (NCEP) Global Forecast System (GFS) soil moisture; (**b**) 1.25 multiples of NCEP GFS soil moisture; (**c**) 1.5 multiples of NCEP GFS soil moisture; (**d**) 2.0 multiples of NCEP GFS soil moisture; (**e**) 2.5 multiples of NCEP GFS soil moisture; and (**f**) 3.0 multiples of NCEP GFS soil moisture.

*2.3. Verification Measurements*

For continuous variables, the verification measures were based on the forecast error (i.e., f-o). To evaluate the continuous forecasts, the bias (BIAS) and root-mean-squared error (RMSE) were used, i.e.,

$$\text{BIAS} = \frac{1}{n} \sum_{i=1}^{n} f_i - o_i \tag{1}$$

$$\text{RMSE} = \sqrt{\frac{1}{n} \sum_{i=1}^{n} f_i - o_i^2} \tag{2}$$

where *f* represents the forecasts, *o* represents the observation, and *n* is the number of forecast-observation pairs. A perfect forecast has BIAS = 0 and RMSE = 0.

The precipitation forecast was evaluated using the critical success index (CSI) and frequency bias (FBIAS). Briefly, CSI is the ratio of the number of times the precipitation was correctly forecasted to occur to the number of times it either was forecasted or occurred. CSI ranges from 0 to 1, and a dperfect forecast would have a CSI value of 1. FBIAS is the ratio of the total number of forecasts of the precipitation to the total number of observations of the precipitation. A "good" value of FBIAS is close to 1; a value greater than 1 indicates that the precipitation was forecasted too frequently; and a value less than 1 indicates that the precipitation was not forecasted frequently enough [31]. The formulations of CSI and FBIAS are given by,

$$\text{TS} = \frac{n_{11}}{n_{11} + n_{10} + n_{01}} \tag{3}$$

$$\text{FBIAS} = \frac{n_{11} + n_{10}}{n_{11} + n_{01}} \tag{4}$$

where $n_{ij}$ represents the counts in each forecast-observation category, *i* represents the forecast, *j* represents the observations, and the two possible forecast and observation values are represented by the values 0 (i.e., miss) and 1 (i.e., hit). Thus, the counts, $n_{11}$, $n_{10}$, $n_{01}$ and $n_{00}$, are sometimes called the "hits", "false alarms", "misses", and "correct rejections", respectively.

## 3. Results and Discussion

*3.1. Impacts of Soil Moisture Initialization on Air Specific Humidity and Potential Temperature*

According to the simulated results of the SCM under different initial conditions of soil moisture content (i.e., 0.1, 0.2, 0.4, 0.6), soil moisture could affect the specific humidity and potential temperature of the boundary layer significantly and rapidly (Figure 3). The results indicate that the specific humidity increases and the potential temperature decreases persistently with the increasing soil moisture at Urumqi (43.79° N, 87.65° E). The horizontal axis represents the time series since 1200 UTC on 15 August, 2019, while the vertical axis shows the model vertical coordinate.

Since the atmosphere is unsaturated (refer to Figure 3), the soil water could absorb sensible heat and radiations and eventually evaporate. Therefore, the atmospheric water vapor increases when soil moisture increases. The increase in specific humidity leads to the drop in the potential temperature. With the implementation of the PBL parameterization scheme in the SCM simulations, water vapor and sensible heat are mostly transported from the ground to the air through eddy diffusion in the atmospheric boundary layer. These simulated results are highly consistent with the observations, i.e., days in the wetter soil period tend to have lower air temperatures, higher humidity and a more stable and shallower BL (as reported by Zhou et al. [3]).

The water vapor evolution of the SCM simulations under different initial conditions of soil moisture content

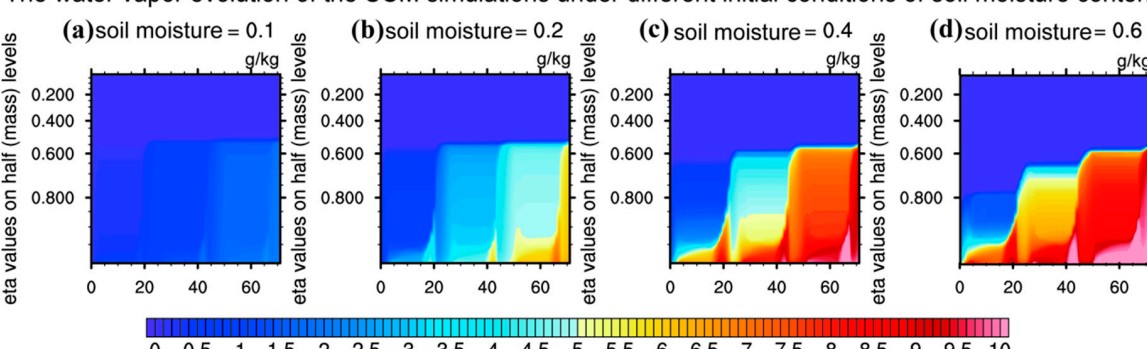

The potential temperature evolution of the SCM simulations under different initial conditions of soil moisture content

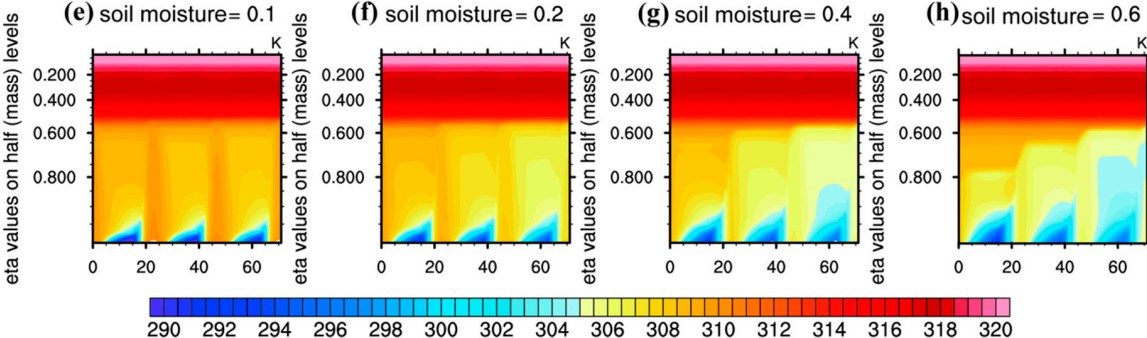

**Figure 3.** The water vapor evolution of the single-column model (SCM) simulations under different initial conditions of soil moisture content: (**a**) soil moisture = 0.1; (**b**) soil moisture = 0.2; (**c**) soil moisture = 0.4; and (**d**) soil moisture = 0.6. The potential temperature evolution of the SCM simulations under different initial conditions of soil moisture content: (**e**) soil moisture = 0.1; (**f**) soil moisture = 0.2; (**g**) soil moisture = 0.4; and (**h**) soil moisture = 0.6. All simulations are at Urumqi (43.79° N, 87.65° E). The *x*-axis represents the hours since the simulation, while the *y*-axis shows the model vertical coordinate (the small value represents high altitude).

### 3.2. Impacts of Soil Moisture Initialization on the Energy Budget at the Surface

Under different soil moisture initial conditions, SCM simulations reveal that the upward moisture fluxes at the ground surface (QFXs) increase when the soil moisture increases (Figure 4d). The increasing QFXs result in the increasing latent heat fluxes and decreasing sensible heat fluxes (Figure 4a,b). Nonetheless, the QFXs, sensible heat fluxes (SHFs) and latent heat fluxes (LHFs) are nearly identical when the atmosphere is almost saturated with water vapor.

Atmospheric water vapor plays an important role in the exchange of energy fluxes between the atmosphere and the land surface [6–9]. The evaporation of soil water absorbs vast heat and radiation, which results in the increase in the LHFs' proportion of total available energy at the land surface. Simultaneously, higher atmospheric water vapor could effectively absorb more long-wave radiation from the ground. This would enhance the greenhouse effect, which would in turn modulate the partition between SHFs, LHFs, upward long-wave radiation fluxes and downward long-wave radiation fluxes at the ground surface.

The total energy of the SHFs and LHFs increases when the soil moisture increases (Figure 4c). This is caused by the enhanced greenhouse effect induced by the increase in atmospheric water vapor. The enhanced greenhouse effect raises the downward long-wave radiation fluxes (GLWs), hence the GLWs increase when the soil moisture increases (Figure 5a). The upward long-wave fluxes at the ground surface (UPLWs) increase in the daytime and decrease in the nighttime as the soil moisture increases (Figure 5d). To understand the reason for this, we studied the formula for calculating UPLWs applied to the WRF model, i.e.,

$$UPLW = \text{Emiss} \times (5.67e - 08) \times TSK^4 + (1 - \text{Emiss}) \times GLW \qquad (5)$$

where "Emiss" represents the ground surface emissivity, and "TSK" represents the surface skin temperature. Following Equation (5), we deduce that the UPLW is a dependent variable for TSK, thus a higher TSK results in more UPLWs. The greenhouse effect of the air moderates the changes in land surface temperature. Hence, the TSK is lower in the daytime and higher in the nighttime under wetter atmospheric conditions. In a sense, an increase in soil moisture induces lower TSK in the daytime, which results in less upward long-wave radiation and induces higher TSK in the nighttime, which causes more upward long-wave radiation.

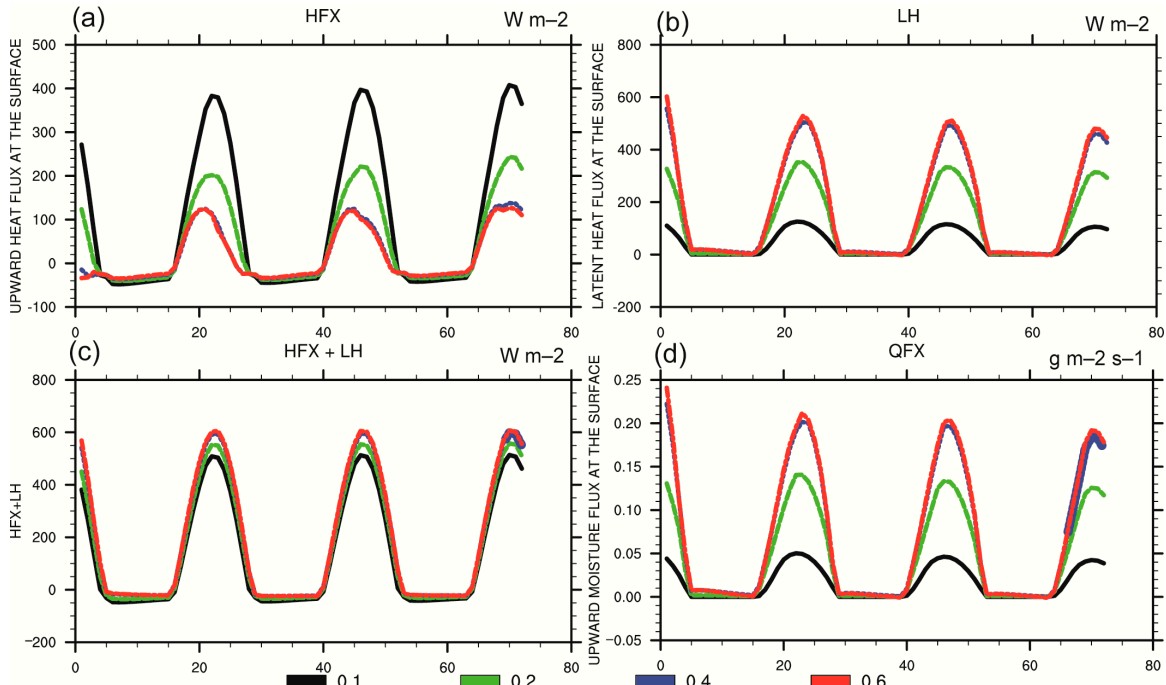

**Figure 4.** The (**a**) sensible heat fluxes (SHFs); (**b**) latent heat fluxes (LHFs); (**c**) LHFs + SHFs and (**d**) upward moisture fluxes at the ground surface (QFXs) of the SCM simulations under different soil moisture initial conditions. The black line represents the soil moisture equal to 0.1; the green line represents the soil moisture equal to 0.2; the blue line represents the soil moisture equal to 0.4; and the red line represents the soil moisture equal to 0.6. The *x*-axis represents the number of hours since the simulation.

There are no significant differences in terms of the downward short-wave radiation (SWDOWN) and upward short-wave radiation (UPSW) at the ground surface (Figure 5b,c) because the atmosphere almost does not absorb short-wave radiation. However, short-wave fluxes of wetter soil diminish marginally because the atmospheric water vapor slightly absorbs short-wave radiation.

The above results reveal that soil moisture could significantly modify the energy balance and modulate the partition between SHFs, LHFs, UPLWs and GLWs at the ground surface through the enhanced greenhouse effect. As the soil moisture increases, the LHFs increase, the SHFs decrease, the sum of SHFs and LHFs increases, the GLWs increase and the UPLWs decrease in the daytime and increase in the nighttime. Nonetheless, these indices would be nearly identical when the atmosphere is nearly saturated.

Based on the observations from the Oasis System Energy and Water Cycle Field Experiment in the Jinta Oasis of Gansu Province, Wen et al. [32] demonstrated that downward short-wave radiation would remain unchanged with modified soil moisture; bigger soil moisture with smaller upward

radiation; and wetter soil, bigger downward long-wave radiation in the daytime. These are more or less consistent with our SCM simulation and verify the rationality of the SCM simulations. However, the SCM simulation provides more comprehensive results of the impacts of soil moisture initialization on the energy budget at the land surface. For instance, the SCM simulation shows that a higher soil moisture leads to a higher upward long-wave radiation at the nighttime.

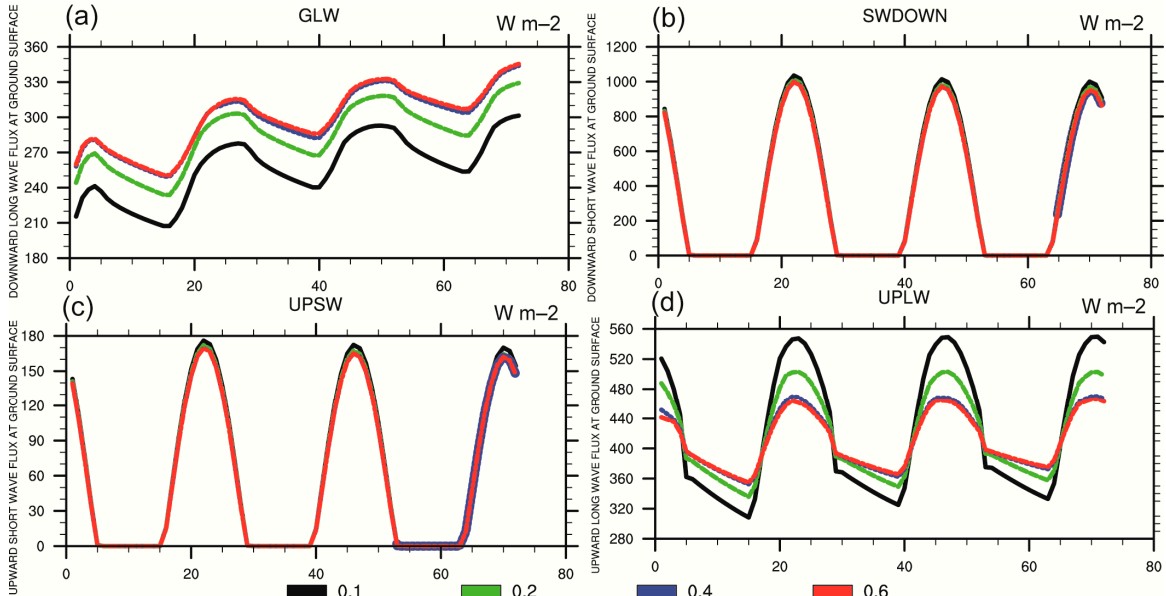

**Figure 5.** The (**a**) downward long-wave radiation fluxes (GLW); (**b**) downward short-wave radiation (SWDOWN); (**c**) upward short-wave radiation (UPSW) and (**d**) upward long-wave radiation (UPLW) of the SCM simulations under different soil moisture initial conditions. The black line represents the soil moisture equal to 0.1; the green line represents the soil moisture equal to 0.2; the blue line represents the soil moisture equal to 0.4; and the red line represents the soil moisture equal to 0.6. The *x*-axis represents the number of hours since the simulation.

### 3.3. Impacts of Soil Moisture Initialization on the Upper Atmospheric Forecasts

Soil moisture initialization could impact the upper air forecasts of WRF significantly and regularly. The characteristics of the spatial distribution of the forecasts of WRF were almost retained invariably when the soil moisture was adjusted by an identical multiple of GFS soil moisture content over all the simulation regions; meanwhile, the intensity of the forecasts of WRF changed significantly and regularly in the same conditions. For example, specific humidity increases beneath the 500-hPa when the soil moisture increases; potential temperature decreases beneath the 500-hPa and increases above the 500-hpa when the soil moisture increases; and geopotential height (GPH) increases beneath 700-hPa and decreases between 400-hPa and 700-hPa as soil moisture increases.

The WRF simulations (see Figure 6) reflect the notion that the specific humidity of 700-hPa increases across most of the simulation regions when soil moisture content increases. However, the spatial pattern of the specific humidity of 700-hPa barely changed (e.g., the dry air in the Taklimakan desert was still drier than other simulation regions). This is probably because the characteristics of the spatial distribution of the forecasts of WRF were almost retained invariably when the soil moisture was adjusted by an identical multiple of GFS soil moisture content for all the simulation regions. Apparently, the intensity of forecasts of WRF changed significantly in the same conditions.

We evaluated the forecasts of upper air against the observations of 14 meteorological sounding stations in Xinjiang, China. However, the upper air sounding data were limited to 0000 and 1200 UTC. To expand the amount of verified forecast-observation pairs, the sounding data from 1200 UTC 15

August to 1200 UTC 18 August 2019 were all used in order to verify the impacts of soil moisture initialization on the WRF forecasts.

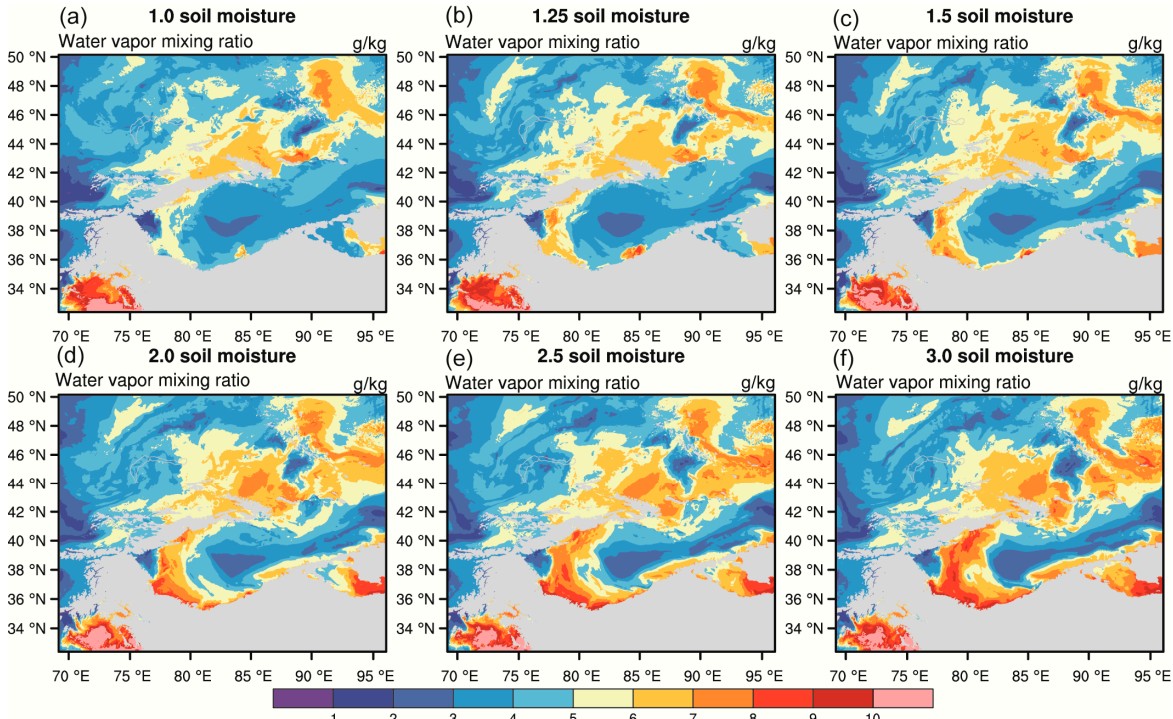

**Figure 6.** The specific humidity of 700-hPa at 002 UTC 17 August 2019 of the WRF simulations under different soil moisture initial conditions. (**a**) 1.0; (**b**) 1.25; (**c**) 1.5; (**d**) 2.0; (**e**) 2.5; and (**f**) 3.0 multiples of NCEP GFS soil moisture represent (**a**) "1.0 soil moisture"; (**b**) "1.25 soil moisture"; (**c**) "1.5 soil moisture"; (**d**) "2.0 soil moisture"; (**e**) "2.5 soil moisture"; and (**f**) "3.0 soil moisture", respectively.

Whilst specific humidity increases beneath 500-hPa when soil moisture increases, it is retained almost invariably under different soil moisture initial conditions above 500-hPa. This might be caused by the extreme scarceness of water vapor above 500-hPa (i.e., above the top of the planetary boundary layer).

Figure 7b demonstrates that the potential temperature decreases beneath 500-hPa and increases above 500-hpa when soil moisture increases. Since water can absorb heat and evaporate into the atmosphere, increasing soil moisture can enhance the atmospheric water vapor and consequently decrease the potential temperature beneath 500-hPa. However, the potential temperature increases above 500-hPa as the soil moisture increases, probably because of the rise in upward long-wave radiation. Further verifications involving observations and simulations would be needed to confirm this.

As shown in Figure 7c, GPH increases beneath 700-hPa and decreases between 400-hPa and 700-hPa as soil moisture increases. In fact, there is a strong correlation between the potential temperature and GPH. A high potential temperature can expand the air volume, reduce air density and decrease GPH. By contrast, a low potential temperature elevates GPH. Therefore, potential temperature increases above 500-hPa can cause GPH to lower between 400-hPa and 700-hPa when soil moisture increases. On the contrary, potential temperature decreases beneath 500-hPa can lift GPH higher beneath 700-hPa as soil moisture increases. Using WRF simulations, Yi et al. [33] suggested that the increase in soil moisture would lead to the decrease of GPH around 750–500 hPa and the increase of GPHs below 850 hPa in Eastern China. Our results are quite consistent with this study, other than the slightly different GPH values.

Figure 7d indicates that soil moisture initialization insignificantly impacts the horizontal wind speed, probably because soil moisture could not directly affect the atmospheric circulation.

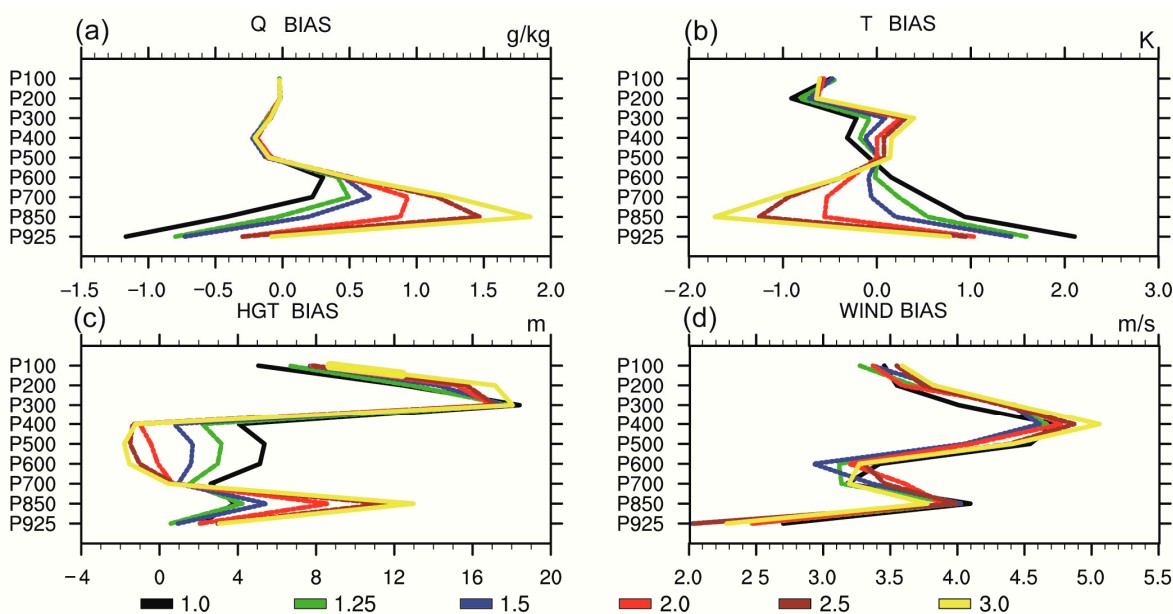

**Figure 7.** (**a**) Specific humidity; (**b**) potential temperature; (**c**) geopotential height (GPH) and (**d**) wind speed BIAS verified against observations of 14 meteorological sounding stations in Xinjiang under different soil moisture initial conditions. Note that the 1.0, 1.25, 1.5, 2.0, 2.5 and 3.0 multiples of NCEP GFS soil moisture are denoted as black, green, blue, red, brown and yellow lines, respectively. The figure shows the verification results for the upper air forecasts. The *x*-axis represents the BIAS, while the *y*-axis represents the values of different isobaric surfaces.

Soil moisture initialization could significantly influence the upper air forecasts of WRF. The characteristics of the spatial distribution of the forecasts of WRF were almost retained invariably when the soil moisture was adjusted by an identical multiple of GFS soil moisture content across all simulation regions. The intensity of the forecasts of WRF changed significantly in the same conditions (e.g., specific humidity increases beneath 500-hPa as soil moisture increases; potential temperature decreases beneath 500-hPa and increases above 500-hpa as soil moisture increases; geopotential height increases beneath 700-hPa and decreases between 400-hPa and 700-hPa as soil moisture increases).

### 3.4. Impacts of Soil Moisture Initialization on the Surface Forecasts

The surface forecasts were assessed against the observed data gathered from 105 surface observation stations in Xinjiang, China.

In the case study, the simulated 2-m specific humidity showed significant dry bias. According to the simulation results under different initial conditions of soil moisture, the 2-m specific humidity increases when soil moisture increases. Hence, the 2-m specific humidity BIAS value changes from negative to positive in Figure 8a.

The simulated 2-m temperature showed significant warm bias in the case study. The 2-m temperature decreases as soil moisture increases, thus the 2-m temperature BIAS value ranged from high to low, as displayed in Figure 7c.

It should be noted that the root-mean-squared errors (RMSEs) of 2-m specific humidity and 2-m temperature are almost minimized (i.e., optimized) when the soil moisture is replaced with 2.0 multiples of the GFS original soil moisture. If the soil moisture continued to increase, the RMSEs would increase (i.e., deteriorate) in Figure 7b,d.

Gómez et al. [34] used the Regional Atmospheric Modeling System (RAMS) to investigate the impact and influence of initial soil moisture distributions on mesoscale circulations in Eastern Spain. They confirmed that high soil moisture is associated with colder near-surface temperatures and a

moister relative humidity, whereas a drier soil would result in a dryer relative humidity and warmer temperature. Our WRF simulations (in Xinjiang, China) are very much in agreement with this study.

Our verification results suggested that soil moisture initialization could significantly impact the surface forecasts (e.g., the 2-m specific humidity increases and the 2-m temperature decreases as soil moisture increases). In some conditions, by replacing the soil moisture content with an appropriate multiple of GFS soil moisture data, the RMSE of 2-m specific humidity and 2-m temperature could be significantly improved.

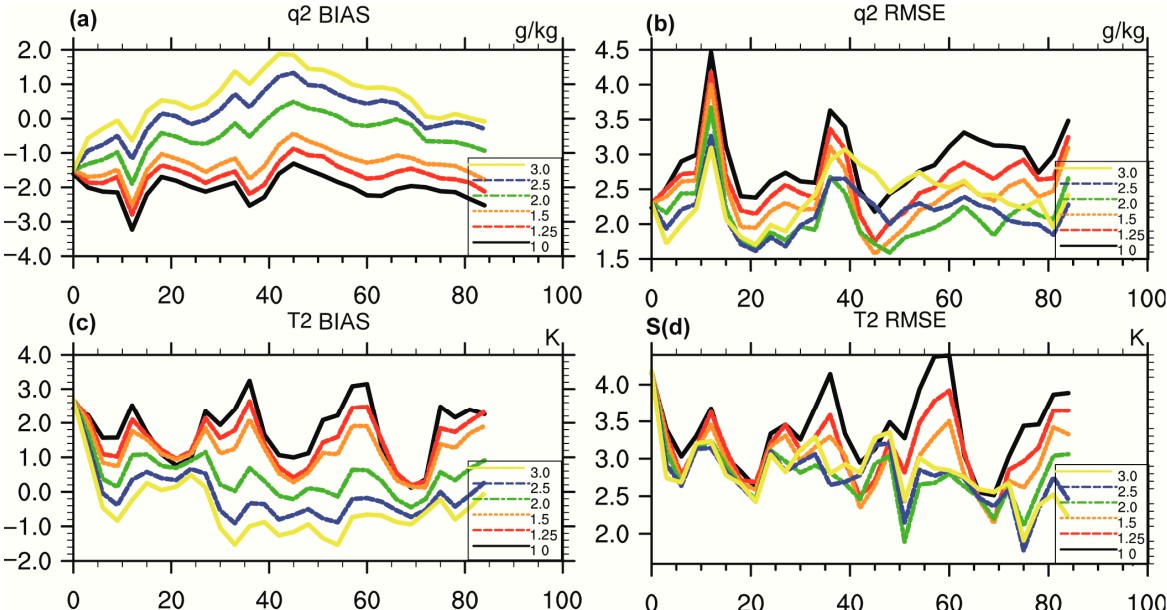

**Figure 8.** The (**a**) 2-m specific humidity BIAS; (**b**) 2-m specific humidity RMSE; (**c**) 2-m temperature BIAS and (**d**) 2-m temperature RMSE verified against observations of 105 meteorological stations in Xinjiang under different soil moisture initial conditions. Note that the 1.0, 1.25, 1.5, 2.0, 2.5 and 3.0 multiples of NCEP GFS soil moisture are denoted as black, red, orange, green, blue and yellow lines, respectively. The figure shows the verification results of the surface forecasts. The *x*-axis represents the hours since the simulation, while the *y*-axis represents the BIAS or RMSE.

### 3.5. Impacts of Soil Moisture Initialization on Precipitation

To evaluate the simulated forecasts with different soil moisture levels, the CSI and FBIAS of the 24-h accumulative precipitation were calculated. To obtain more reliable results, all forecast-observation pairs of 105 meteorological stations at every simulated valid time were used in the calculation.

The characteristics of rainfall spatial distribution simulated under different soil moisture conditions suggest that the intensity of precipitation is more sensitive to the soil moisture content than the characteristics of the spatial distribution of the precipitation (e.g., the precipitation center remains nearly unchanged with different soil moisture content levels). The intensity of precipitation increases significantly when soil moisture increases, as shown in Figure 9.

According to the simulation results of our verification, soil moisture initialization can impact precipitation regularly and significantly. By replacing the soil moisture content with an appropriate multiple of the GFS soil moisture content across all simulation regions, CSI and FBIAS are significantly improved. Nevertheless, either excessive or scarce soil water moisture content would worsen the CSI and FBIAS of precipitation.

In the studied weather case, the simulated precipitation shows significant underestimation (see Figure 10g–j). Therefore, the 24-h accumulative precipitation FBIAS is lower than one at some magnitudes (from 0.1 mm to 12.1 mm), e.g., the FBIAS of 0.1 mm is 0.81 and that of 12.1 mm is 0.51. This indicates that the atmospheric water vapor is drier in the model than in the realistic conditions.

By contrast, the FBIAS of precipitation greater than 24.1 mm shows significant overestimation because the number of forecast-observation pairs ranges (only) from 40 to 60. This might be caused by model errors or perhaps stochastic issues due to limited samples (where new investigations would be required). Nevertheless, we excluded the 24-h accumulative precipitation values equal or greater than 24.1 mm in the subsequent analysis.

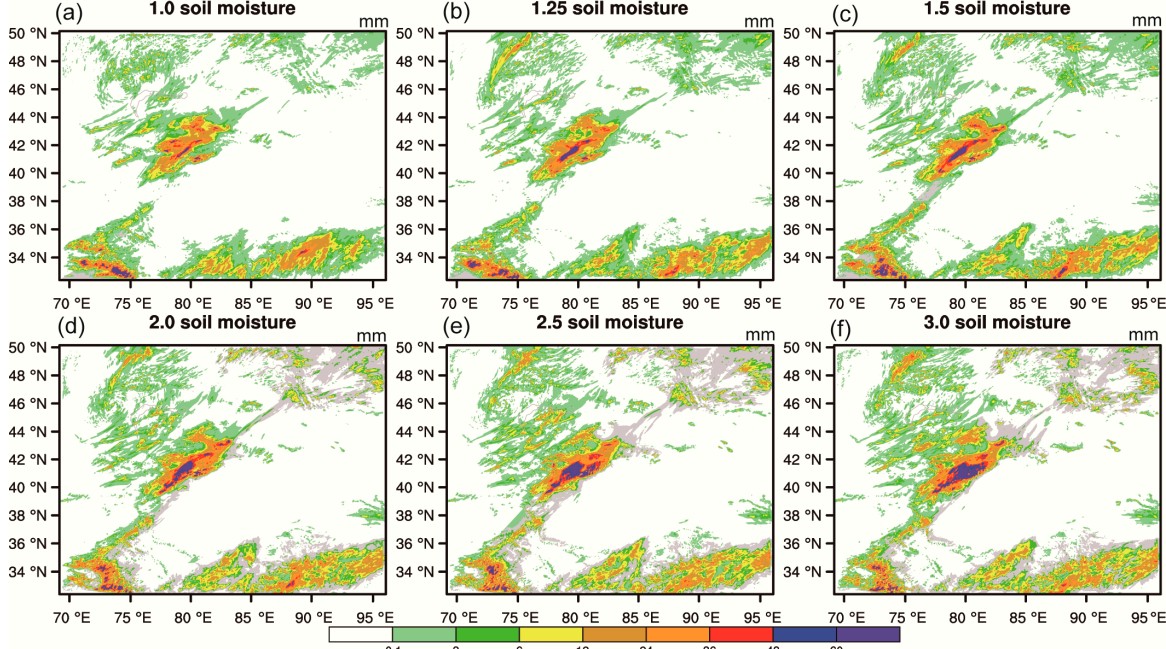

**Figure 9.** The 24-h accumulative precipitation at 002 UTC 17 August 2019 of the WRF simulations under different soil moisture initial conditions. Note that (**a**) 1.0; (b) 1.25; (**c**) 1.5; (**d**) 2.0; (**e**) 2.5; and (**f**) 3.0 multiples of NCEP GFS soil moisture are denoted as (**a**) "1.0 soil moisture"; (**b**) "1.25 soil moisture"; (**c**) "1.5 soil moisture"; (**d**) "2.0 soil moisture"; (**e**) "2.5 soil moisture"; and (**f**) "3.0 soil moisture", respectively. Precipitation is measured in millimeters.

The dry bias of the atmosphere in the model might be caused by the dry bias of the soil moisture in accordance with the tests under different initial conditions of soil moisture content. As soil moisture increases, the CSI and FBIAS of 24-h accumulative precipitation improve significantly under the conditions in which soil moisture content is equal to or less than 1.5 multiples of GFS soil moisture content. When soil moisture content exceeds 1.5 multiples, CSI or FBIAS would worsen at some magnitude. Considering all CSI and FBIAS at different magnitudes of 24-h accumulative precipitation, the optimum value of soil moisture is 2.0 multiples of GFS soil moisture content.

It should be noted that, in the current case study, increasing soil moisture tends to improve CSI and FBIAS. However, other case studies might require researchers to decrease the soil moisture. This depends on whether the initial conditions of a weather simulation case show wet bias or dry bias relative to the real situation. Based on the WRF model, Vivoni et al. [35] examined the effects of initial soil moisture on rainfall generation in the Upper Río Puerco (URP) basin in New Mexico and suggested that the total rainfall, intensity and spatial coverage increase with higher soil moisture. The simulations here are generally consistent with this study.

In summary, soil moisture could affect the precipitation regularly and significantly. CSI and FBIAS improved significantly when the soil moisture content was set to an appropriate multiple of the GFS soil moisture content across all simulation regions. Nonetheless, either excessive or scarce soil water moisture content would worsen the CSI and FBIAS of precipitation.

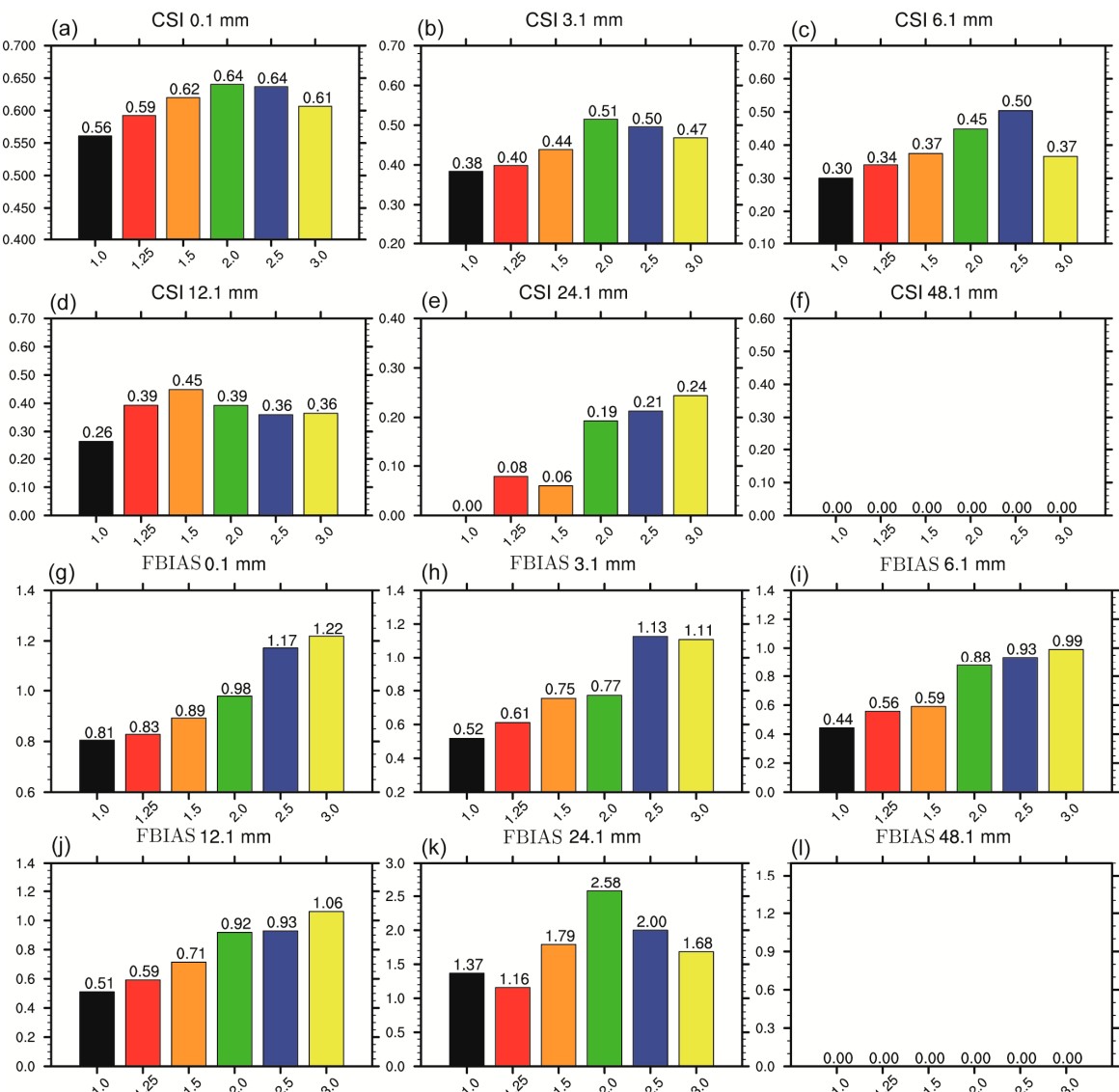

**Figure 10.** The critical success index (CSI) and frequency bias (FBIAS) values of 24-h accumulative precipitation of the WRF simulations under different soil moisture initial conditions at each precipitation magnitude: (**a,g**) 0.1 mm; (**b,h**) 3.1 mm; (**c,i**) 6.1 mm; (**d,j**) 12.1 mm; (**e,k**) 24.1 mm; and (**f,l**) 48.1 mm. The 1.0, 1.25, 1.5, 2.0, 2.5 and 3.0 multiples of NCEP GFS soil moisture are represented as black, red, orange, green, blue and yellow columns, respectively.

## 4. Conclusions and Future Studies

In this paper, we rigorously investigated the impacts of soil moisture initialization on the forecasts of the WRF model. The findings of this study are as follows: (1) significant and regular relationships exist between soil moisture initial conditions at different magnitudes and the forecasts of WRF (including the upper air and surface forecasts); (2) the atmospheric greenhouse effect (which is regulated by soil moisture) plays an important role in the budget and partition of the total available energy at the land surface in the WRF model; (3) replacing the soil moisture with a proper multiple of the NCEP GFS soil moisture data could significantly improve the accuracy of the forecasts of the WRF model. Furthermore, we used the SCM to isolate the simulations from the interference of the large-scale advections in order to obtain reliable and robust results of soil moisture impacts on the forecasts of the WRF model.

These are new findings which were rarely reported in previous studies. Whilst some studies mentioned the impacts of soil moisture on the energy balance at the land surface through evaporation [6–9], they failed to clarify how soil moisture regulated the greenhouse effect.

This study should provide a more comprehensive and profound understanding of the soil moisture initialization impact on the forecasts of the WRF model and provide prior knowledge for the effective application of WRF in Xinjiang, China or other middle-high latitude regions with similar climate conditions. In particular, the prospect of replacing the soil moisture with a proper multiple of the NCEP GFS soil moisture data (item (3) in this section) is practical and the accuracy of the forecasts of the WRF model should be improved. It would be interesting to figure out how to obtain the value of the proper multiple of the NCEP GFS soil moisture in future studies.

**Author Contributions:** Conceptualization, H.Z. and X.M.; data curation, X.M. and A.A.; formal analysis, H.Z. and H.L.; funding acquisition, H.L.; investigation, H.Z., X.M. and A.A.; methodology, H.Z., J.L. and H.L.; resources, J.L., H.L. and A.A.; software, H.Z., J.L. and A.A.; supervision, X.M. and H.L.; validation, H.Z., J.L. and A.A.; visualization, J.L.; writing—original draft, H.Z.; writing—review & editing, H.Z. All authors have read and agreed to the published version of the manuscript.

**Funding:** This research was funded by the Central Scientific Research Institute of the Public Basic Scientific Research Business Professional (Grant No. IDM2017001), the National Natural Science Foundation of China (Grant No. 41875023), the Research Foundation of China Desert Meteorology (Grant No. sqj2018017), the National Department of Public Benefit (Meteorology) Research Foundation (Grant No. GYHY201306066), the Central Asia Atmospheric Research Foundation (Grant No. CASS201711).

**Conflicts of Interest:** The authors declare no conflict of interest.

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
