# Peer review of "The Impacts of Soil Moisture Initialization on the Forecasts of Weather Research and Forecasting Model: A Case Study in Xinjiang, China"

_water, doi:10.3390/w12071892_

Round 1

Reviewer 1 Report

The article presents an interesting and important subject of the impacts of soil moisture initialization on the forecasts of the Weather Research and Forecasting model. It should be noted that the article is based on actual literature related to analyzed subject. The manuscript is well-written. The authors put effort into explaining the research results clearly - in an exhaustive way.

Only a few minor comments are noted – listed below:

  1. I suggest shortening the abstract – it is a little too long. The abstract should present concisely the research problems and objectives, the brief description of the methods used, only key results and conclusions.
  2. Line 41: 7 cites per sentence are presented, it should be specified what is their contribution and why are they mentioned in the paper (what these references have done).
  3. Lines 197 and 200: If I am not wrong, this is the first time the acronym „LHFs” and „SHFs” are mentioned in the text (apart from abstract). I suggest putting the full form in the first place and the acronym in brackets.
  4. Figures 4, 5, 6, 7, 8, 9, 10: above the figures, there are given the same titles as given in the figure captions. Please remove the titles above the figures – they are not needed.
  5. Line 246: „(...) and regularly in the same conditions, .” – please remove the comma.
  6. Figure 7: Please check the vertical axis of the first graph (a) – there is „missing” instead of P100.
  7. Figure 8: The legends may be slightly reduced to fit into the graphs.
  8. Figure 10: The caption may be shortened – two sentences are almost identical, it would be good to combine them into one, shorter sentence.

Reviewer 2 Report

This paper focused on “The Impacts of Soil Moisture Initialization on the Forecasts of WRF: a Case Study in Xinjiang, China” The research topic of this paper is interesting and has contribution to the academic community, but there are serious deficiencies in current manuscript which needs to be made major revision to reach the publishing level.

  1. Following are the comments/recommendations, the authors should consider seriously. 
  2. Title and highlights: Abbreviations should not be in the title and highlights
  3. The authors should read the abstract carefully for major corrections of English language.
    Significant improvements are needed in the introduction. This part is too verbose and there is too much irrelevant information. It is suggested to consider merging with the literature review.
  4. literature review part is too poor to simply list the previous research results, but it needs to correspond with the research content of this paper and point out the shortcomings of the previous research results so as to clarify the significance of this paper.
  5. Authors should concentrate on literature review and justification of study. 
  6. Need focus on data and method section… for detail check review version of MS
  7. Results should be well justified and discussion section should be focused and concise.
    The discussion section should be well based on defending of results wile comparing with other studies.
  8. Revised conclusion section, here no need to explain the results in the conclusion and policy recommendation section. It should be focused to conclude and draw recommendations based on results.
  9. This article needs native English polishing.
    In a word, this paper needs to be made major revision before it is published.

Reviewer 3 Report

The paper shows results of research simulations of finding the impacts of soil moisture initialization on the forecasts of Weather Research and Forecasting. The investigations were planned properly, course of estimation is clear and results are satisfactory. In the Introduction, in my opinion, the sentence "The remainder of this paper is organized as follows. Section 2 briefly introduces the detailed designation, configuration and data of experiments. All results are described and discussed in section 3. Conclusions and further studies are presented in section 4." should be removed. Readers can find the structure of the paper while reading. The little lack of the paper is too many promises of the future studies (especially in the Conclusions). This raises the question why this has not been done so far? I suggest to limit the promises.

Additionally, the paper should be edited carefully (uniform spaces between lines, distance between figures and text, all references should be citied as a number in square bracket, not as "Hong et al. (2009)").

Summarize, in my opinion, the paper brings some novelty, overall scientific soundness is average with high significant of the content. After minor revision it could be published. The theme is very important in the view of global climate changes. Some practical recommendation how to balance the soil moisture to achieve the optimal temperature could be a good conclusion what could make the paper more worthy.

Reviewer 4 Report

In this paper the authors show a very interesting study on soil moisture initialisation on the forecasts of Weather Research and Forecasting (WRF) model. Two groups of simulations were carried, on the one hand the SCM was used to simulate ideal cases and, on the other hand, the WRF model was used to analyse real cases. The authors use the effect of the moisture initialisation by means of several statistics. The conclusion is that soil moisture initialization could affect the 24 forecasts of WRF significantly and regularly.

The paper is very interesting, but, in my humble opinion, some aspects can be improved:

  1. The quality of some figures can be improved. For example, labels of figure 2 are too small; figure 3 is not clear; figures 7 and 8 are hard to understand…
  2. In figure 4: results for 0.4 and 0.6 are equivalent, is this correct?
  3. The authors propose a “bias-correction” of soil moisture by multiplying the GFS soil moisture to improve the regional numerical model forecasting performance: I think this point is not clear.
  4. The authors say in line 353 “This might be caused by model errors and perhaps stochastic issue due to limited samples.” This point need more investigation.

Following these comments my decision is that this work cannot be published as it is.

Round 2

Reviewer 2 Report

Thanks for your kind efforts and hard work to improve your MS. I accepting MS.

Congratulation